# Effect of Ultraviolet–Ozone Treatment on the Properties and Antibacterial Activity of Zinc Oxide Sol-Gel Film

**DOI:** 10.3390/ma12152422

**Published:** 2019-07-29

**Authors:** Ji-Hyeon Kim, Junfei Ma, Seunghun Lee, Sungjin Jo, Chang Su Kim

**Affiliations:** 1Advanced Nano-Surface Department, Korea Institute of Materials Science, 797, Changwon-daero, Sungsan-gu, Changwon 51508, Korea; 2School of Architectural, Civil, Environmental, and Energy Engineering, Kyungpook National University, Daegu 41566, Korea

**Keywords:** zinc oxide, antibacterial film, sol-gel film, UV–ozone effect, film surface investigation

## Abstract

To combat infectious diseases, zinc oxide (ZnO) has been identified as an effective antibacterial agent; however, its performance can be adversely affected by harsh application environments. The ozone impact on ZnO antibacterial film needs to be evaluated prior to its application in an ozone disinfection system. In this study, ZnO films synthesized via sol-gel/spin-coating were subjected to ultraviolet–ozone (UVO) treatment for different periods. Surface investigations using scanning electron microscopy, ultraviolet–visible spectroscopy, and X-ray photoelectron spectroscopy revealed that the treatment-induced film changes. With longer UVO treatment, the surface porosity of the film gradually increased from 5% to 30%, causing the transmittance reduction and absorbance increase in visible-light range. Phase transformation of Zn(OH)_2_ to ZnO occurred during the first 10 min of UVO treatment, followed by oxygen uptake as a consequence of the reaction with reactive oxygen species generated during UVO treatment. However, despite these surface changes, the satisfactory antibacterial activity of the synthesized ZnO film against *Staphylococcus aureus* and *Escherichia coli* was sustained even after 120 min of UVO treatment. This indicates that the UVO-induced surface changes do not have a significant effect on the antibacterial performance and that the ZnO sol-gel film possesses good functional durability in ozone environments.

## 1. Introduction

Today, infectious diseases as well as environmental pollution have drawn attention worldwide. Particularly, hospital-acquired infections (called nosocomial infections) that occur in the hospital or other healthcare locations are considered an inevitable challenge for public health [1,2,3,4,5,6,7]. Nosocomial infections are estimated to annually result in at least 138,000 deaths worldwide, according to a report by the World Health Organization [8]. Accordingly, for the last two decades, antibacterial coatings along with antibiotics have been developed to prevent nosocomial infections and subsequent mortality. As major antibacterial agents depress the proliferation, spread, and infection of diverse strains, metal oxides such as ZnO [9,10,11,12,13,14,15,16,17], CuO [13,16,18], TiO_2_ [19], Fe_2_O_3_ [13], Ag_2_O [20], CaO [11], and MgO [11,21,22] have been extensively researched.

Among the metal-oxide agents, zinc oxide (ZnO) has been recognized as one of the promising antibacterial agents thanks to its inherent characteristics, such as good biocompatibility with humans (non-toxicity), biological effectiveness for bacteria-cell killing, low cost, and excellent physicochemical and thermal resistance—enabling its longer lifetime and higher cost-effectiveness than other organic and inorganic materials [12,23,24,25,26,27,28,29,30]. It has been proposed that the antibacterial effect of ZnO is attributed to its three major antibacterial mechanisms: (1) formation of reactive oxygen species (ROS) [9,11,25]; (2) liberation of antibacterial ions—mainly Zn^2+^ ions [30,31,32]; and (3) internalization into the cell wall of strains, destroying the structural and functional integrity of the cell [23,33,34].

It has been experimentally verified that the performance of the ZnO antibacterial material is considerably affected by a variety of parameters, for example, its composition and structural properties (thickness for film structure, mechanical size for particle form, morphology, porosity, surface defects, and impurities) associated with the specific surface area [9,10,12,16,24,25,28,29,35,36,37]. These parameters can be influenced by the coating method. To synthesize high-quality ZnO films, many fabrication methods such as sol–gel [38,39,40,41], atomic layer deposition [42,43], spray pyrolysis [44,45], reactive or non-reactive magnetron sputtering [46,47,48], reactive electron beam evaporation [49], chemical vapor deposition [50,51], laser or plasma molecular-beam epitaxy [52,53,54,55], and electrodeposition [56] have been employed. Sol–gel, in particular, is a potential method with several advantages such as simple and low-cost processing, low processing temperatures, great purity and super-uniformity of the films, and controllability for the size and shape of sol particles [40,41]. Above all, sol-gel has two unique strong-points; one is a top-down process enabling the production of the size and shape of particles, and the other is the capability of coating the diverse applications with non-flat surfaces. These abilities are essential to achieve the acceptable performance of antibacterial films.

As well as the fabrication method of antibacterial films, the surrounding environment can affect their properties, and hence functional performance. In particular, a harsh application environment, —such as corrosive, wearable, frictional, and detachable conditions—can degrade the antimicrobial properties of the applied antibacterial film and even make them lose their functionality. Therefore, the functional durability of antibacterial films is required for their reliability and long lifetime (i.e., economic efficiency) in a practical environment.

Currently, it is being attempted to synergistically enhance the sterilization performance of ozone disinfection systems using antibacterial films in the systems. Ozone disinfectors are commercially used due to their safe, fast, and effective sterilization. An ozone sterilizer produces ozone gas through the following common process: oxygen gas is injected into the system chamber and then subjected to an electrical discharge, which decomposes oxygen molecules into monatomic oxygen and then reforms into ozone. The generated ozone—as a strong oxidizer—quickly kills even tough and resistant pathogenic bacteria. Simultaneously, the ozone providing oxidative stress can degrade or even lose functionality in the applied antibacterial films. Accordingly, it should first be determined whether the performance of the antibacterial film is sustainable in ozone conditions. Unfortunately, there are very few reports describing the ozone effect on antibacterial films [57,58].

In this paper, as a function of UVO exposure time, ZnO antibacterial film prepared by a simple spin-coating method was investigated with respect to the stability in the surface morphology, transmittance/absorbance, chemical state, and antibacterial activity by using field emission scanning electron microscopy (FE-SEM), ultraviolet–visible spectroscopy (UV-Vis), and X-ray photoelectron spectroscopy (XPS).

## 2. Materials and Methods

### 2.1. Preparation of the ZnO Film

ZnO films were prepared by a commonly-used sol-gel method reported in [59,60,61,62], as illustrated in Figure 1. ZnO solution was fabricated by dissolving 1.64 g zinc acetate dihydrate (Zn(CH_3_COO)_2_·2H_2_O) in the mixed solution of 10 g 2-methoxyethanol (CH_3_OC_2_H_5_OH) and 0.5 g mono-ethanolamine (HOCH_2_CH_2_NH_2_, MEA). The MEA was utilized as an organic stabilizer. The zinc acetate dihydrate was completely dissolved via 20-min ultrasonic vibration until no sediment was visible. Then, the transparent solution was filtered using a 0.45-μm membrane filter to obtain a ZnO precursor sol-type solution without any precipitates. To synthesize a uniform ZnO film, the precursor solution was spin-coated on a glass at 1500 revolutions per minute (rpm) for 30 s. The used glass had been ultrasonically cleaned in a glass container of acetone and isopropyl alcohol for 10 min, respectively, and was successively dried at 150 °C for 1 min to volatilize the solvent and organic residuals.

### 2.2. UV–Ozone Treatment

For the prepared ZnO films, the UVO treatment of 0, 10, 30, 60, and 120 min was carried out using a commercial UVO cleaner (YOC130326, YUILUV co., Incheon, South Korea) containing UV light sources, a chamber, and injection and exhaust lines. The used UV light source was 185/254 nm from 150 W mercury lamps. The prepared ZnO samples were placed at a position of approximately 2 cm from the UV lamps. At that distance, the intensity of the generated UV light reaches 40 mW/cm^2^. Hereafter, the synthesized ZnO film is referred to as “ZnO,” and the UVO-treated ZnO films are denoted as “treatment time UVO ZnO” (e.g., 10-min UVO ZnO).

### 2.3. Characterization

The thickness and surface morphology of the ZnO samples were analyzed via a stylus profilometer (DektakXT, Bruker co., Billerica, Germany) and a SEM (JSM-7610F, JEOL co., Akishima, Japan), respectively. To investigate the chemical state, an XPS (K-Alpha, Thermo Fisher co., Waltham, USA) consisting of an X-ray source of 1.5 keV, an Al Kα monochromator, and a 128-channel detector was used. The transmission/absorption analysis was conducted using a UV-Vis (Cary 5000 UV-Vis-NIR, Agilent co., Santa Clara, CA, USA). The blank baseline of the UV-Vis had been set with air, and then the transmission and absorption spectrum were recorded in the wavelength range of 200 to 800 nm.

### 2.4. Antibacterial Test

According to the Japanese Industrial Standard (JIS) Z 2801 [63], the antibacterial activity of the prepared and UVO-treated ZnO films against *Staphylococcus aureus* ATCC 6538P (*S. aureus*) and *Escherichia coli* ATCC 8739 (*E. coli*) were evaluated. In brief, 0.1–0.4 ml of a bacterial culture of 10^5^ to 10^6^ colony-forming units per ml (CFU/ml) was injected onto a test plane of a 5 × 5 cm sample. The test plane was covered with a sterilized ethylene film of 4 × 4 cm and successively incubated in a petri dish at (35 ± 1) °C for 2, 4, or 24 h under a humid environment of 90% relative humidity (to prevent sterilization by desiccation). Immediately after the incubation, the test plane and cover film were sufficiently washed off with 10 ml SCDLP broth medium (i.e., an extraction solution). Exactly 1 ml of the washing liquid taken by a measuring pipette was injected in a test tube containing 9 ml phosphate-buffered saline and then sufficiently mixed by shaking the tube. The resulting dilution was spread in a sterile petri dish and subsequently intermixed with 15–20 ml plate count agar warmed up to 46–48 °C. After the complete solidification of the culture medium, the petri dish was turned upside down and incubated in an incubation chamber at (35 ± 1) °C for 40–48 h. Right after the incubation, the bacterial colonies in each petri dish were counted visually. To improve the reliability, this antibacterial test was performed in triplicate for each sample.

## 3. Results and Discussion

### 3.1. Morphological, Optical, and Chemical Properties

As identified by a stylus-profilometry measurement and SEM surface low-magnification image analysis, the synthesized ZnO film showed 36-nm thickness and surface continuity without micro-scaled agglomeration. After UVO treatment, its surface continuity was sustained, while the nanoscale surface morphology changed, as presented in the SEM high-magnification top-view image of Figure 2a. As can be seen in Figure 2a,b, the pores over the ZnO film surface decreased in size but increased in density and quantity as the UVO treatment proceeded. The surface porosity values, marked as points in the graph of Figure 2b, were obtained by calculating the pore area fraction of each Figure 2a image using the widely-used image analysis program, “ImagJ”. With longer UVO treatment, the ZnO sol-gel film possessed a higher surface porosity and, hence, greater bactericidal activity. Some recent publications [10,14,15] have revealed experimentally that the functional performance of ZnO antibacterial coating against various bacterial strains is in proportion to the specific surface area, which is because of its surface-dependent antibacterial mechanisms mentioned in the introduction.

In addition to the antibacterial activity, optical properties can be affected by the surface structure. As displayed in Figure 2c, the synthesized ZnO film only absorbed UV light and had good transmittance of visible light and near infrared wavelengths; however, all the UVO ZnO samples displayed broad optical absorption, extending to the near infrared region of 800 nm. This optical change was consistent with the visual observation that the transparent synthesized ZnO film became slightly opaque after UVO exposure. In general, porous or rough surfaces tend to cause scattering of incident visible light, leading to a reduction in the transmittance and an increase in the probability of light absorption [64]. It has also demonstrated experimentally that a porous ZnO film prepared by the sol-gel/dipping method showed a low transmittance in the visible light wavelength region, in comparison with a compact ZnO film [65]. Based on the absorbance-transmittance equation (i.e., A = 2 − log_10_ % T, where A is absorbance and %T is the transmittance percentage), the ZnO porous film can be deduced to show a greater absorbance. In addition, the experimental observation, which showed that the absorbance of the ZnO film in the UV wavelength slightly increased with the increasing UVO treatment time is in good agreement with the experimental results on nanostructured ZnO thin film by Ghamsari et al. [39].

As seen in Figure 3a Zn-O phase can be indirectly identified from the Zn and O peaks. ZnO (zinc monoxide) with a high thermodynamic stability is believed to be formed, which was experimentally demonstrated by X-ray diffraction analysis in a research [61] where ZnO film were fabricated in almost the same manner as in this study. Figure 3b,c shows the high-resolution spectrum of the ZnO and UVO ZnO samples in the energy range of the O 1s core-level, revealing the chemical composition and chemical state of the O element present within the sample surface (usually the top 10 nm based on the inherently limited analysis-depth of XPS). The energy scale for all the XPS spectrums was properly calibrated by fixing the C 1s level into 284.8 eV. As can be confirmed in Figure 3a, the O 1s peak can be divided into three peaks centered at 530.3, 531.6, and 532.2 eV. The main O 1s peak shifted to a lower binding-energy position from 532.2 to 531.6 eV, via a 10-min UVO treatment. After 30 min of UVO processing, a shoulder peak at 530.3 eV appeared for the first time. The smallest energy level peak at 530.3 eV (denoted as Zn-OH in Figure 3) is assigned to the oxygen atoms in zinc hydroxide (Zn(OH)_2_) [66,67,68], and the other peaks at 531.6 (Zn-O) and 532.2 eV (V_O_) are associated with the oxygen atoms in ZnO with and without oxygen vacancies, respectively [69,70,71,72]. Hence, it can be interpreted that the Zn(OH)_2_ in the synthesized ZnO sample was fully transformed into ZnO with some oxygen vacancies during the first 10-min UVO treatment, and afterwards the oxygen vacancies were filled with oxygen atoms (i.e., oxygen uptake occurred). As identified from the peak-fitting results of Figure 3b, 24.2% of the oxygen vacancies were filled throughout 30-min UVO treatment. The experimental peak was deconvoluted by fitting the peak with a mixed Gaussian–Lorentz function.

In reality, ZnO films synthesized by using a Zn(CH_3_COO)_2_·2H_2_O precursor (as in this study) can contain some Zn(OH)_2_, which can be acceptable when considering the synthesis-associated chemical chain reactions [38,69,73,74,75,76,77]:Zn(CH_3_COO)_2_∙2H_2_O → Zn(CH_3_COO)_2_ + 2H_2_O(1)
Zn(CH_3_COO)_2_ + 2H_2_O → 2CH_3_COOH + Zn(OH)_2_(2)
Zn(OH)_2_ + 4NH_2_CH_2_CH_2_OH → [Zn(NH_2_CH_2_CH_2_OH)_4_]^2+^ + 2OH^–^(3)
Zn(OH)_2_ + 2OH^–^ → [Zn(OH)_4_]^2–^(4)
Zn(OH)_2_ → ZnO + H_2_O(5)

Zn(OH)_2_ can be generated as an intermediate in the course of the synthesis. From the thermogravimetric and differential thermal analysis (TG/DTA) result reported in [78], the thermal decomposition from Zn(OH)_2_ to ZnO and water (Equation (5)) took place at a temperature range of 110–140 °C. However, in the case of the short (1 min) drying time at 150 °C in this study, it is suspected that the Zn(OH)_2_ present in the spin-coated ZnO film was not totally decomposed to ZnO throughout the drying. Moreover, according to the documented TG/DTA results of ZnO sample [79,80], it was observed that the solvent (i.e., 2-methoxyethanol) and water remaining over the ZnO sample volatilized at 66 and 133 °C, respectively. On the other hand, structural decomposition of the hydroxyl group (i.e., Zn(OH)_2_) occurred at a relatively higher temperature, 297 °C. Consequently, it is expected that some Zn(OH)_2_ remained in the surface and/or inside of the synthesized ZnO film in this study. This is in agreement with other experimental results of ZnO sol-gel films fabricated by using a Zn(CH_3_COO)_2_·2H_2_O precursor at a low processing temperature [59,81].

As depicted in Figure 3a, the Zn(OH)_2_ was transformed into ZnO during UVO treatment, which is presumed to enhance the antibacterial activity. Fiedot et al. [82] reported that ZnO microrods deposited on PET mesh by a chemical bath deposition method contained a different amount of Zn(OH)_2_ depending on the deposition temperature. The higher the ratio of ZnO to Zn(OH)_2_, the greater the antimicrobial activity against *E. coli*, *S. aureus*, *Staphylococcus epidermidis*, and *Candida albicans*.

Additionally, the oxidation (more precisely, oxygen uptake) of the ZnO film in the UVO process can be explained. Once compressed air is supplied into the chamber of the used UVO cleaner, atomic oxygen is dissociated by 185 nm UV radiation, and, afterwards, ozone is continuously produced by 254 nm radiation [83,84,85,86,87,88], as follows:O_2_ + hv(185 nm) → O(3p) + O(3p),(6)
O(3p) + O_2_ + O_2_ → O_3_ + O_2,_(7)
O_3_ + hv(254 nm) → O(1d) + O_2._(8)

As can be seen in Equations (6) and (8), ozone (O_3_) and oxygen free radicals (O(3p) and O(1d)), which are a type of ROS (i.e., strong oxidants), were believed to be generated during UVO treatment. Thus, it is predicted that the resulting ROS caused the oxidation phenomena (i.e., oxygen uptake) of the ZnO film.

### 3.2. Bactericidal Effectiveness

In order to evaluate the antibacterial activity of the prepared ZnO sol-gel film as a function of the contact time and UVO treatment time, antibacterial tests were carried out according to the JIS Z 2801 standard [63]. The test organisms used in this study were *S. aureus* and *E. coli*, which are gram-positive and gram-negative bacterial strains, respectively. To achieve repeatability and reproducibility, each antibacterial test was validated in accordance with the guideline documented in the JIS Z 2801 standard [63].

Figure 4a,b and Table 1 show the test results of the synthesized ZnO film. The control sample (i.e., bare glass without ZnO film) showed that some of the test strain was killed up to the first 4-h incubation time; however, the test strain proliferated after 24 h. Compared to the control, the synthesized ZnO film exhibited superior antibacterial activity, killing 99.8% or 99.9% (4-log reduction for *S. aureus* and 6-log reduction for *E. coli*) of both test bacterial strains—irrespective of the incubation time. Given the antibacterial test results of other metal-oxide agents such as ZnO_2_ [89], SiO_2_ [90], and TiO_2_ [91], the ZnO sol-gel film in this paper is estimated to be an excellent agent. Furthermore, satisfactory antibacterial activity of the synthesized ZnO sol-gel film was maintained even after the 120-min UVO treatment, as displayed in Figure 4c. Thus, it can be concluded that the performance sustainability of the ZnO film is excellent under ozone environments.

## 4. Conclusions

Our study experimentally demonstrated that a ZnO film prepared by a sol-gel/spin-coating method morphologically, optically, and chemically changed as a function of the UVO treatment time. However, the antibacterial ability of the film was sustained. The key findings and conclusions are as follows:As the UVO treatment progressed, the film surface showed a decrease in the porosity size and an enormous increase in the porosity density, eventually resulting in an increase in the surface porosity and, thus, the specific surface area. The resulting surface, which was more porous, induced lower transmittance and higher absorbance in the visible light region.O 1s XPS spectrum analysis revealed that the chemical state of the film surface also changed via UVO treatment. Firstly, the Zn(OH)_2_ present in the prepared ZnO sol-gel film was transformed into ZnO containing oxygen vacancies. Afterwards, the oxygen vacancies were partially occupied with oxygen atoms as a consequence of ROS generation during UVO treatment. That is, a solid–solid transformation (or a dehydration reaction) and an oxidation reaction sequentially occurred on the ZnO film surface.However, the UVO treatment-induced changes over the ZnO film surface did not degrade its antibacterial effectiveness against *S. aureus* and *E. coli*. Rather, the porosification and Zn(OH)_2_ to ZnO transformation induced by the UVO treatment improved the bactericidal ability due to the specific surface area increase and the great antibacterial activity of ZnO being superior to that of Zn(OH)_2._.

Given these experimental results, it can be deduced that the ZnO sol-gel antibacterial film is a potential way to synergistically improve commercially-used ozone disinfection systems for a long service time.

## Figures and Tables

**Figure 1 materials-12-02422-f001:**
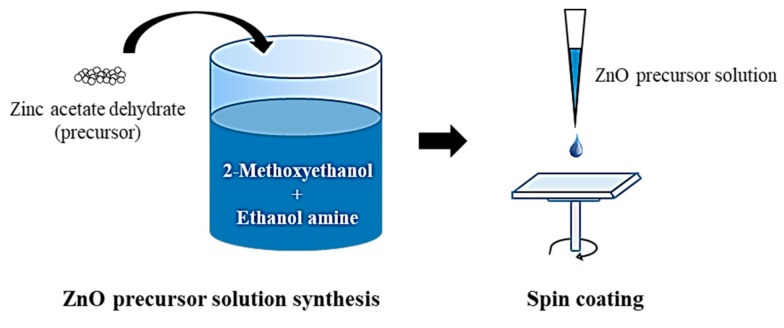
Schematic representation of the sol-gel fabrication process used to obtain the ZnO thin film.

**Figure 2 materials-12-02422-f002:**
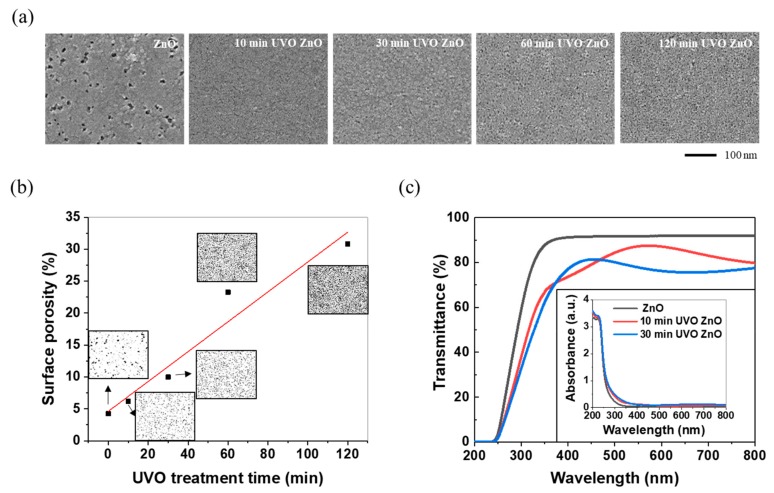
(**a**) SEM top-view images in secondary-electron image mode, (**b**) SEM image-based surface porosity, and (**c**) UV-Vis transmittance and absorbance spectrum of ZnO and UVO ZnO samples.

**Figure 3 materials-12-02422-f003:**
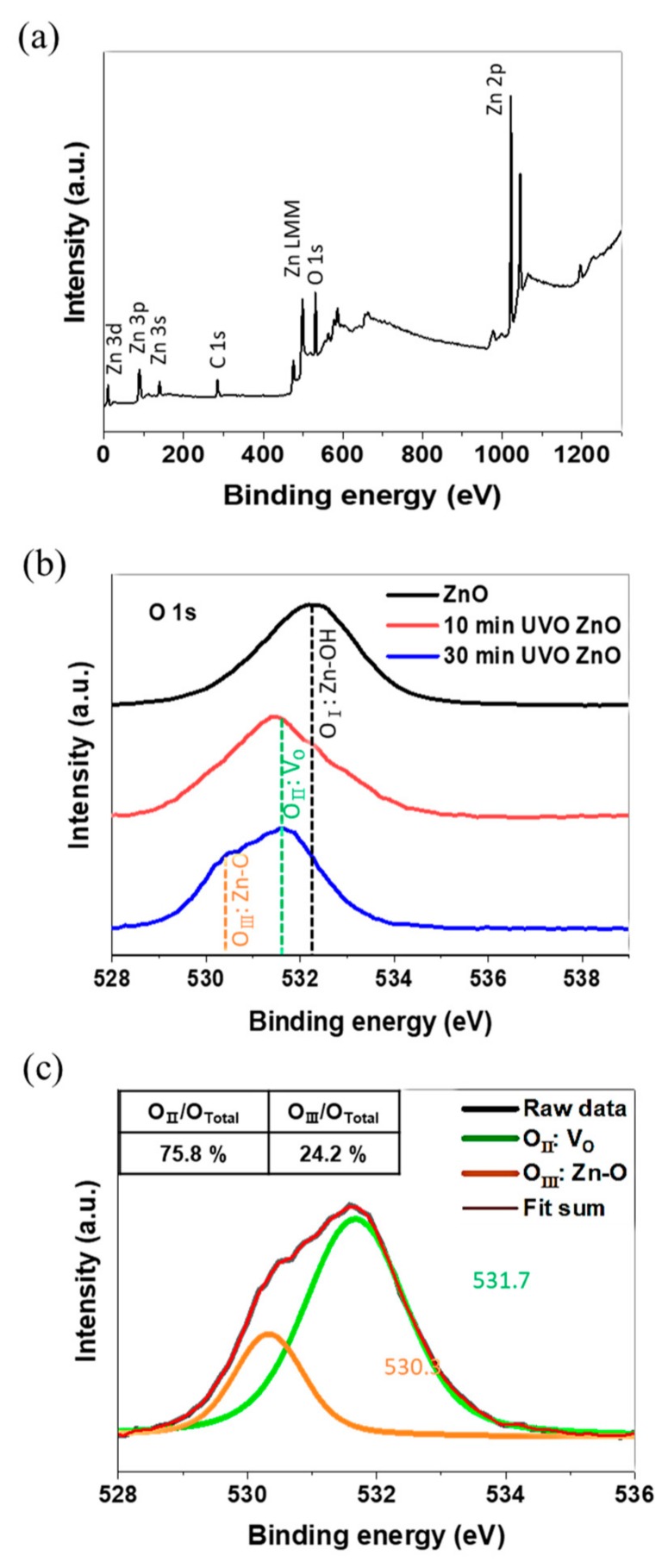
Baseline-corrected high-resolution XPS spectrum: (**a**) wide-scan spectrum of the 30-min UVO ZnO film, (**b**) comparison among ZnO and UVO ZnO films at the O 1s energy level, and (**c**) the high-resolution deconvoluted O 1s spectrum of the 30-min UVO ZnO film.

**Figure 4 materials-12-02422-f004:**
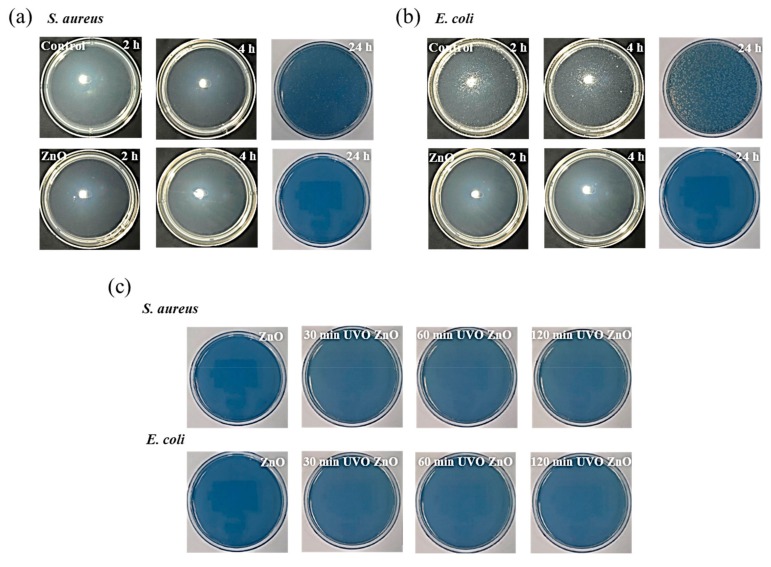
Antibacterial test (JIS Z 2801) results of the prepared ZnO film against *S. aureus* (gram-negative bacteria) and *E. coli* (gram-negative bacteria) for ZnO film, in terms of the (**a**,**b**) contact time and (**c**) UVO exposure time (the contact time was fixed as 24 h).

**Table 1 materials-12-02422-t001:** Antibacterial activity of ZnO film as a function of the UVO treatment time, test bacteria, and contact time.

	Percent reduction (%)					
Test Organism	*S. aureus*			*E. coli*		
Contact Time	2 h	4 h	24 h	2 h	4 h	24 h
ZnO	99.9	99.9	99.9	99.8	99.9	99.9
30-min UVO ZnO	–	–	99.9	–	–	99.9
60-min UVO ZnO	–	–	99.9	–	–	99.9
120-min UVO ZnO	–	–	99.8	–	–	99.9
Control	2.0	0.6	−41.2	6.0	52.0	−7547.1

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
