# Peer review of "Effect of Ultraviolet–Ozone Treatment on the Properties and Antibacterial Activity of Zinc Oxide Sol-Gel Film"

_materials, 2019, doi:10.3390/ma12152422_

Round 1
Reviewer 1 Report
This study is original, the experiments are performed and analyzed with care, using several complementary morphological, chemical and optical spectroscopy techniques. The manuscript is well organized and written.
For all the above reasons, I recommend this paper for publication, provided that the following points are addressed in a revised version (minor revision requested):
1. The main drawback of the manuscript is that there is no structural evidence for the presence of ZnO as main crystallographic phase in the film, although the film is declared as ZnO. Usually, in the sol-gel film deposition from a zinc acetate precursor, a higher temperature is needed to decompose the carbonic material, eg. above 300 C, see the recent article J. Thermal Anal. Calorimetry 135, 2019, pag 2943 and cited references. If it is not possible to perform structural analysis in due time (X-ray or electron diffraction), at least some more arguments for the presence of ZnO should be presented. There is some evidence by XPS, but the film is thicker than the 10 nm investigated by XPS and other chemical elements from precursors were not evaluated, only the oxygen presence.
2. Page 2, line 51: please be more specific about the terms “size” and “concentration”. For example do you mean the “nano-grain size”?
3. Page 2, Section 2.1: As the preparation method is not original, please mention several related references, eg. Appl. Surf. Sci. 396, 2017, Pag. 1880, J. Thermal Anal. Calorimetry 135, 2019, pag 2943 and cited references.
4. Page 3, figure 1: I think you should specify “ZnO precursor solution” instead of “ZnO solution” along with the right side drawing?
5. Page 5, paragraph starting with line 153: references are missing in order to sustain/support the assignations made by XPS and presented in figure 3.
6. Page 6, paragraph starting with line 181: For the thermal decomposition of Zn(OH)2 into ZnO, could you add some references about the temperature range transformation? I suggest J. Alloys Compd. 548, 2013, pag 222. Regarding the discussion of the thermal decomposition of the precursor to ZnO in the sol-gel film, please add references related to temperatures in the synthesis and phase transformation. This request is related in fact with the first observation (point 1).
Author Response
To reviewer,
First of all, thank you very much for your constructive comments and suggestions that allowed us to greatly improve the quality of the manuscript. We agree with all your comments, and we corrected point by point the manuscript accordingly. Your comments are in bold text and our responses in plain italics.
Received Comments and Suggestions
1. The main drawback of the manuscript is that there is no structural evidence for the presence of ZnO as main crystallographic phase in the film, although the film is declared as ZnO. Usually, in the sol-gel film deposition from a zinc acetate precursor, a higher temperature is needed to decompose the carbonic material, eg. above 300 C, see the recent article J. Thermal Anal. Calorimetry 135, 2019, pag 2943 and cited references. If it is not possible to perform structural analysis in due time (X-ray or electron diffraction), at least some more arguments for the presence of ZnO should be presented. There is some evidence by XPS, but the film is thicker than the 10 nm investigated by XPS and other chemical elements from precursors were not evaluated, only the oxygen presence.
Revision: As you suggested, ZnO crystallographic phase of the film could be indirectly confirmed from the XPS wide-scan spectra ranging from 0 to 1350 eV binding energy, together with a supportive reference. The result has been inserted in line 166–169 (Page 6) and Figure 3a(Page 6), as follows;
“As seen in Figure 3a, Zn-O phase can be indirectly identified from the Zn and O peaks. ZnO (zinc monoxide) with a high thermodynamic stability is believed to be formed, which was experimentally demonstrated by X-ray diffraction analysis in a research [61] where ZnO film were fabricated in almost the same manner as in this study.”
Figure 3. Baseline-corrected high-resolution XPS spectrum: (a) wide-scan spectrum of the 30-min UVO ZnO film, (b) comparison among ZnO and UVO ZnO films at the O 1s energy level, and (c) the high-resolution deconvoluted O 1s spectrum of the 30-min UVO ZnO film.
2. Page 2, line 51: please be more specific about the terms “size” and “concentration”. For example, do you mean the “nano-grain size”?
Revision: The vague or unsuitable words have been changed or removed accordingly in page 2, line 52 of the revised paper, as follows:
“thickness for film structure, mechanical size for particle form”
3. Page 2, Section 2.1: As the preparation method is not original, please mention several related references, eg. Appl. Surf. Sci. 396, 2017, Pag. 1880, J. Thermal Anal. Calorimetry 135, 2019, pag 2943 and cited references.
Revision: I have added the associated references including your recommended references in page 2, line 87 of the revised manuscript.
4. Page 3, figure 1: I think you should specify “ZnO precursor solution” instead of “ZnO solution” along with the right side drawing?
Revision: I have revised the naming accordingly in page 3, Figure 1 of the revised manuscript.
5. Page 5, paragraph starting with line 153: references are missing in order to sustain/support the assignations made by XPS and presented in figure 3.
Revision: The related references has been added in page 5, sentence starting with line 176–178 of the revised manuscript, as follows:
“The smallest energy level peak at 530.3 eV (denoted as Zn-OH in Figure 3) is assigned to the oxygen atoms in zinc hydroxide (Zn(OH)2) [66–68], and the other peaks at 531.6 (Zn-O) and 532.2 eV (VO) are associated with the oxygen atoms in ZnO without and with oxygen vacancies, respectively [69–72].”
6. Page 6, paragraph starting with line 181: For the thermal decomposition of Zn(OH)2 into ZnO, could you add some references about the temperature range transformation? I suggest J. Alloys Compd. 548, 2013, pag 222. Regarding the discussion of the thermal decomposition of the precursor to ZnO in the sol-gel film, please add references related to temperatures in the synthesis and phase transformation. This request is related in fact with the first observation (point 1).
Revision: The supplementary contents you mentioned have been added accordingly in page 5, line 193–204 of the revised paper, as follows:
“From the thermogravimetric and differential thermal analysis (TG/DTA) result reported in [78], the thermal decomposition from Zn(OH)2 to ZnO and water (Equation (5)) took place at a temperature range of 110–140 °C. However, in the case of the short (1-min) drying time at 150 °C in this study, it is suspected that the Zn(OH)2 present in the spin-coated ZnO film was not totally decomposed to ZnO throughout the drying. Moreover, according to the documented TG/DTA results of ZnO sample [79,80], it was observed that the solvent (i.e., 2-methoxyethanol) and water remaining over the ZnO sample volatilized at 66 and 133 °C, respectively. On the other hand, structural decomposition of the hydroxyl group (i.e., Zn(OH)2) occurred at a relatively higher temperature, 297 °C. Consequently, it is expected that some Zn(OH)2 remained in the surface and/or inside of the synthesized ZnO film in this study. This is in agreement with other experimental results of ZnO sol-gel films fabricated by using a Zn(CH3COO)2·2H2O precursor at a low processing temperature [61,81].”

Reviewer 2 Report
-The authors should reworded the abstract section in order to highlight the focus of research.
-it is possible to obtain a roughness of the surface with AFM?
-the conclusions not supported very well the work. please reword.
Author Response
To reviewer,
First of all, thank you for your constructive comments and suggestions that allowed us to greatly improve the quality of the manuscript. We agree with all your comments, and we corrected point by point the manuscript accordingly. Your comments are in bold text and our responses in plain italics.
Received Comments and Suggestions
1. The authors should reworded the abstract section in order to highlight the focus of research.
Revision: The abstract has been revised accordingly. Please see the abstract in page 1 of the revised paper.
2. it is possible to obtain a roughness of the surface with AFM?
Revision: In our study, we focused on the pores in the surface structure of the ZnO films. The resolution can be not good if the probe is not sharp enough. In the case of the ZnO films, normal AFM were not enough to measure the pore depth (i.e., roughness). This is because the pores are as small as the error in the width measurement by AFM (usually, 5–10 nm).
3. the conclusions not supported very well the work. please reword.
Revision: The conclusion has been revised accordingly. Please see the conclusions in page 8 of the revised paper.

Reviewer 3 Report
In the present work the ZnO films synthesized by sol-gel / spin-coating have been subjected to ultraviolet-ozone (UVO) treatment for different periods of time. The materials were characterized using scanning electron microscopy (SEM), ultraviolet-visible, spectroscopy (UV-Vis) and X-ray photoelectron spectroscopy (XPS). Furthermore, it has been shown that despite ultraviolet-ozone (UVO) treatment, the antibacterial activity against two different strains has not changed
The manuscript addresses an interesting topic and describes clearly the experimental methods and results.
Moreover, the paper also uses a good English. Therefore, it may be recommended for publication after minor revision:
1. Please, the authors should better describe the sol-gel method and why they use this synthesis method in the Introduction section.
2. Please, the authors should add some references to reinforce the antibacterial results obtained: Coatings, 2019, 9(3),200; Materials, 2019, 12(1),155
Author Response
To reviewer,
First of all, thank you for your constructive comments and suggestions that allowed us to greatly improve the quality of the manuscript. We agree with all your comments, and we corrected point by point the manuscript accordingly. Your comments are in bold text and our responses in plain italics.
Received Comments and Suggestions
1. Please, the authors should better describe the sol-gel method and why they use this synthesis method in the Introduction section.
Revision: The related contents as written below has been contained in page 2, line 54–63 of the revised manuscript, as follows:
“To synthesize high-quality ZnO films, many fabrication methods such as sol–gel [38–41], atomic layer deposition [42,43], spray pyrolysis [44,45], reactive or non-reactive magnetron sputtering [46–48], reactive electron beam evaporation [49], chemical vapor deposition [50,51], laser or plasma molecular-beam epitaxy [52–55], and electrodeposition [56] have been employed. Specially, sol–gel is a potential method with several advantages such as simple and low-cost processing, low processing temperatures, great purity and super-uniformity of the films, and controllability for the size and shape of sol particles [40,41]. Above all, sol-gel has two unique strong-points; one is a top-down process enabling to produce the size and shape of particles, and the other is the capability to coat the diverse applications with non-flat surfaces. These abilities are essential to achieve the acceptable performance of antibacterial films.”
2. Please, the authors should add some references to reinforce the antibacterial results obtained: Coatings, 2019, 9(3),200; Materials, 2019, 12(1),155
Revision: Several references including the references you recommended have been added accordingly in page 7, line 239–240 of the revised paper, as follows:
“Given the antibacterial test results of other metal-oxide agents such as ZnO2 [89], SiO2 [90], and TiO2 [91], the ZnO sol-gel film in this paper is estimated to be an excellent agent.”
